# Generating Liquid Simulations with Deformation-aware Neural Networks

**Lukas Prantl, Boris Bonev & Nils Thuerey**
Department of Computer Science
Technical University of Munich
Boltzmannstr. 3, 85748 Garching, Germany
{lukas.prantl,boris.bonev,nils.thuerey}@tum.de

## Abstract

We propose a novel approach for deformation-aware neural networks that learn the weighting and synthesis of dense volumetric deformation fields. Our method specifically targets the space-time representation of physical surfaces from liquid simulations. Liquids exhibit highly complex, non-linear behavior under changing simulation conditions such as different initial conditions. Our algorithm captures these complex phenomena in two stages: a first neural network computes a weighting function for a set of pre-computed deformations, while a second network directly generates a deformation field for refining the surface. Key for successful training runs in this setting is a suitable loss function that encodes the effect of the deformations, and a robust calculation of the corresponding gradients. To demonstrate the effectiveness of our approach, we showcase our method with several complex examples of flowing liquids with topology changes. Our representation makes it possible to rapidly generate the desired implicit surfaces. We have implemented a mobile application to demonstrate that real-time interactions with complex liquid effects are possible with our approach.

## 1 Introduction

Learning physical functions is an area of growing interest within the research community, with applications ranging from physical priors for computer vision problems Kyriazis & Argyros (2013), over robotic control Schenck & Fox (2017), to fast approximations for numerical solvers Tompson et al. (2017). While the underlying model equations for many physics problems are known, finding solutions is often prohibitively expensive for phenomena on human scales. At the same time, the availability of model equations allows for the creation of reliable ground truth data for training, if enough computational resources can be allocated.

Water, and liquids in general, are ubiquitous in our world. At the same time, they represent an especially tough class of physics problems, as the constantly changing boundary conditions at the liquid-gas interface result in a complex space of surface motions and configurations. In this work we present a novel approach to capture parametrized spaces of liquid behavior that is based on space-time deformations. We represent a single 3D input surface over time as a four-dimensional signed-distance function (SDF), which we deform in both space and time with learned deformations to recover the desired physical behavior. To calculate and represent these deformations efficiently, we take a two-stage approach: First, we span the sides of the original parameter region with pre-computed deformations, and infer a suitable weighting function. In a second step, we synthesize a dense deformation field for refinement. As both the parameter weighting problem and the deformation synthesis are highly non-linear problems, we demonstrate that neural networks are a particularly suitable solver to robustly find solutions.

We will demonstrate that it is possible to incorporate the non-linear effects of weighted deformations into the loss functions of neural networks. In particular, we put emphasis on incorporating the influence of deformation alignment into the loss gradients. This alignment step is necessary to ensure the correct application of multiple consecutive deformations fields. The second stage of our

algorithm is a generative model for deformation fields, for which we rely on a known parametrization of the inputs. Thus, in contrast to other generative models which learn to represent unknown parametrization of data sets Radford et al. (2016), our models are trained with a known range and dimensionality to parameter range, which serves as input.

Once trained, the models can be evaluated very efficiently to synthesize new implicit surface configurations. To demonstrate its performance, we have implemented a proof-of-concept version for mobile devices, and a demo app is available for Android devices in the *Google Play store*. Our approach generates liquid animations several orders of magnitude faster than a traditional simulator, and achieves effective speed up factors of more than 2000, as we will outline in Sec. 5. The central contributions of our work are:

- A novel *deformation-aware neural network approach* to very efficiently represent large collections of space-time surfaces with complex behavior.

- We show how to compute suitable loss gradient approximations for the sub-problems of parameter and deformation inference.

- In addition we showcase the high performance of our approach with a mobile device implementation that generates liquid simulations interactively.

## 2 RELATED WORK

Capturing physical behavior with learning has a long history in the field of learning. Early examples targeted minimization problems to determine physical trajectories or contact forces Bhat et al. (2002); Brubaker et al. (2009), or plausible physics for interacting objects Kyriazis & Argyros (2013; 2014). Since initial experiments with physically-based animation and neural networks Grzeszczuk et al. (1998), a variety of new deep learning based works have been proposed to learn physical models from videos Battaglia et al. (2016); Chang et al. (2016); Watters et al. (2017). Others have targeted this goal in specific settings such as robotic interactions Finn et al. (2016), sliding and colliding objects Wu et al. (2015; 2016), billiard Fragkiadaki et al. (2015), or trajectories in height fields Ehrhardt et al. (2017). The prediction of forces to infer image-space motions has likewise been targeted Mottaghi et al. (2016a;b), while other researchers demonstrated that the stability of stacked objects can be determined from single images Lerer et al. (2016); Li et al. (2016). In addition, the unsupervised inference of physical functions poses interesting problems Stewart & Ermon (2017). While these methods have achieved impressive results, an important difference to our method is that we omit the projection to images. I.e., we directly work with three-dimensional data sets over time.

In the context of robotic control, physics play a particularly important role, and learning object properties from poking motions Agrawal et al. (2016), or interactions with liquids Schenck & Fox (2017) were targeted in previous work. Learning physical principles was also demonstrated for automated experiments with reinforcement learning Denil et al. (2016). Recently, first works have also addressed replacing numerical solvers with trained models for more generic PDEs Farimani et al. (2017); Long et al. (2017). In our work we target a more narrow case: that of surfaces deforming based on physical principles. However, thanks to this narrow scope and our specialized loss functions we can generate very complex physical effects in 3D plus time.

Our method can be seen as a generative approach representing samples from a chaotic process. In this context, the learned latent-space representation of regular generative models Masci et al. (2011); Rezende et al. (2014); Goodfellow et al. (2014); Radford et al. (2016); Isola et al. (2017) is replaced by the chosen parametrization of a numerical solver. Our model shares the goal to learn flow physics based on examples with other methods Ladicky et al. (2015); Tompson et al. (2017); Chu & Thuerey (2017), but in contrast to these we focus on 4D volumes of physics data, instead of localized windows at single instances of time. Alternative methods have been proposed to work with 4D simulation data Raveendran et al. (2014); Thuerey (2017), however, without being able to capture larger spaces of physical behavior. Due to its focus on deformations, our work also shares similarities with methods for optical flow inference and image correspondences Bailer et al. (2016); Dosovitskiy et al. (2015); Ranjan & Black (2016); Ilg et al. (2016). A difference to these approaches is that we learn deformations and their weighting in an unsupervised manner, without explicit ground truth data. Thus, our method shares similarities with spatial transformer networks (STNs) Jaderberg et al. (2015), and unsupervised approaches for optical flow Meister et al. (2017).

However, instead of aligning two data sets, our method aims for representing larger, parametrized spaces of deformations.

# 3    LEARNING DEFORMATIONS

We first explain our formulation of the simulation parameter space, which replaces the latent space of other generative models such as autoencoders Masci et al. (2011), GANs Radford et al. (2016), or auto-regressive models Van Oord et al. (2016). Given a Navier-Stokes boundary value problem with a liquid-gas interface, we treat the interface over time as a single space-time surface. We work with a set of these space-time surfaces, defined over a chosen region of the N-dimensional simulation parameters $\boldsymbol{\alpha}$. We assume the parameters to be normalized, i.e. $\boldsymbol{\alpha} \in [0, 1]^N$. In practice, $\boldsymbol{\alpha}$ could contain any set of parameters of the simulation, e.g. initial positions, or even physical parameters such as viscosity. We choose implicit functions $\phi(\boldsymbol{\alpha}) \in \mathrm{R}^4 \to \mathrm{R}$ to represent specific instances, such that $\Gamma(\boldsymbol{\alpha}) = \{\mathbf{x} \in \mathrm{R}^4; \phi(\boldsymbol{\alpha}, \mathbf{x}) = 0\}$ is the space of surfaces parametrized by $\boldsymbol{\alpha}$ that our generative model should capture. In the following, $\phi$ and $\psi$ will denote four-dimensional signed distance functions. We will typically abbreviate $\phi(\boldsymbol{\alpha})$ with $\phi_{\boldsymbol{\alpha}}$ to indicate that $\phi_{\boldsymbol{\alpha}}$ represents a set of constant reference surfaces. While we will later on discretize all functions and operators on regular Cartesian grids, we will first show the continuous formulation in the following.

**Deforming Implicit Surfaces**    Representing the whole set $\phi_{\boldsymbol{\alpha}}$ is very challenging. Due to bifurcations and discretization artifacts $\phi_{\boldsymbol{\alpha}}$ represents a process that exhibits noisy and chaotic behavior, an example for which can be found in Fig. 1. Despite the complex changes in the data, our goal is to find a manageable and reduced representation. Thus, we generate an approximation of $\phi_{\boldsymbol{\alpha}}$ by deforming a single initial surface in space and time. We apply space-time deformations $\mathbf{u} : \mathrm{R}^4 \to \mathrm{R}^4$ to the initial surface $\psi_0(\mathbf{x})$. As a single deformation is limited in terms of its representative power to produce different shapes, we make use of $N$ sequential, pre-computed deformations, thus $\mathbf{u}_i$ with $i \in [1 \cdots N]$, each of which is scaled by a scalar weight parameter $\beta_i$, whose values are normally between 0 and 1. This gives us the freedom to choose how much of $\mathbf{u}_i$ to apply. The initial surface deformed by, e.g., $\mathbf{u}_1$ is given by $\psi_0(\mathbf{x} - \beta_1 \mathbf{u}_1)$.

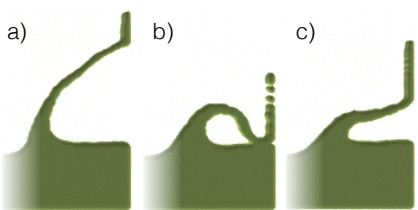

Figure 1: Three liquid surfaces after 60 time steps differing only by $\pm\epsilon$ in initial conditions. Even this initially very small difference can lead to large differences in surface position, e.g., the sheet in b) strongly curving downward.

The sequence of deformed surfaces by sequentially applying all pre-computed $\mathbf{u}_i$ is given by $\psi_i(\mathbf{x}, \boldsymbol{\beta}) = \psi_{i-1}(\mathbf{x} - \beta_i \mathbf{u}_i)$. It is crucial to align such sequences of Eulerian deformations with each other. Here we employ the alignment from previous work Thuerey (2017), which we briefly summarize in the following, as it influences the gradient calculation below. Each deformation $i$ relies on a certain spatial configuration for the input surface from deformation $i-1$. Thus, when applying $\mathbf{u}_{i-1}$ with a weight $\beta_{i-1} < 1$, we have to align $\mathbf{u}_i$ correspondingly. Given the combined deformation $\mathbf{v}_{\mathrm{sum}}(\mathbf{x}, \boldsymbol{\alpha}) = \sum_{i=1}^{N} \beta_i \mathbf{u}_i^*(\mathbf{x})$, with intermediate deformation fields $\mathbf{u}_{i-1}^*(\mathbf{x}) = \mathbf{u}_{i-1}(\mathbf{x} - \mathbf{u}_i^*(\mathbf{x}))$, we compute an inversely weighted offset field as $\mathbf{v}_{\mathrm{inv}}(\mathbf{x}, \boldsymbol{\alpha}) = -\sum_{i=1}^{N}(1 - \beta_i) \mathbf{u}_i^*(\mathbf{x})$. This offset field is used to align the accumulated deformations to compute the final deformation as $\mathbf{v}_{\mathrm{fin}}(\mathbf{x} + \mathbf{v}_{\mathrm{inv}}(\mathbf{x}, \boldsymbol{\beta}), \boldsymbol{\beta}) = \mathbf{v}_{\mathrm{sum}}(\mathbf{x}, \boldsymbol{\beta})$. $\mathbf{v}_{\mathrm{fin}}$ now represents all weighted deformations $\beta_i \mathbf{u}_i$ merged into a single vector field. Intuitively, this process moves all deformation vectors to the right location for the initial surface $\psi_0$, such that they can be weighted and accumulated.

To achieve the goal of representing the full set of target implicit surfaces $\phi_{\boldsymbol{\alpha}}$ our goal is to compute two functions: the first one aims for an optimal weighting for each of the deformations in the sequence, i.e. $\boldsymbol{\beta}$, while the second function computes a final refinement deformation $\mathbf{w}$ after the weighted sequence has been applied. We will employ two neural networks to approximate the two functions, which we will denote as $f_p$, and $f_d$ below. Both functions depend only on the simulation parameters space $\boldsymbol{\alpha}$, i.e., $f_p(\boldsymbol{\alpha}) = \boldsymbol{\beta}$, and $f_d(\boldsymbol{\alpha}) = \mathbf{w}$.

Splitting the problem into $f_p$ and $f_d$ is important, as each of the pre-computed deformations weighted by $f_p$ only covers a single trajectory in the space of deformed surfaces. In the follow-

Figure 2: This illustration gives an overview of our algorithm. It works in two stages, a weighting and refinement stage, each of which employs a neural network to infer a weighting function and a dense deformation field, respectively.

ing, we employ an optical flow solve from previous work to pre-compute deformations between the inputs Thuerey (2017), which deform the input for the extrema of each original parameter dimension $\alpha_i$. E.g., for a two-dimensional parameter space this yields two deformations along the sides of the unit cube. This first step robustly covers rough, large scale deformations. As a second step, we employ $f_d$, which is realized as a generative CNN, to infer a deformation for refining the solution. Below we will explain the resulting equations for training. The full equations for applying the deformations, and a full derivation of our loss functions can be found in the supplemental materials. To shorten the notation, we introduce the helper function $\mathcal{D}(\mathbf{x}_i, \boldsymbol{\alpha})$, which yields a deformed set of coordinates in $\mathrm{R}^4$ depending on $\boldsymbol{\alpha}$ that incorporates the deformation sequence weighted by $f_p(\boldsymbol{\alpha})$, and a refinement deformation from $f_d(\boldsymbol{\alpha})$.

We express the overall goal in terms of minimizing the $L_2$ distance between the deformed and the target implicit surfaces for all possible values in the parameter space $\boldsymbol{\alpha}$, using $\boldsymbol{\beta}$ and $\mathbf{w}$ as degrees of freedom:

$$\underset{\boldsymbol{\beta}, \mathbf{w}}{\operatorname{argmin}} \, L, L = \int \|\psi_0(\mathcal{D}(\mathbf{x}_i, \boldsymbol{\alpha})) - \phi_{\boldsymbol{\alpha}}\|_2^2 \, \mathrm{d}\alpha \,. \tag{1}$$

Our work addresses the problem of how to compute weighting of the deformations and on synthesizing the refinement field. The main difficulty lies in the non-linearity of the deformations, which is why we propose a novel method to robustly approximate both functions with NNs: $f_p$ will be represented by the *parameter network* to compute $\boldsymbol{\beta}$, and we make use of a *deformation network* that to generate $\mathbf{w}$. We employ relatively simple neural networks for both functions. Key for training them is encoding the effect of deformations in the loss functions to make the training process aware of their influence. Hence, we will focus on describing the loss functions for both networks and the corresponding discretized gradients in the following.

**Learning Deformation Weights** For training the NNs we propose the following objective function, which measures the similarity of a known reference surface $\phi_{\boldsymbol{\alpha}}$ and the corresponding, approximated result $\psi_0(\mathbf{x}, \boldsymbol{\alpha})$ for a parameter value $\boldsymbol{\alpha}$. We introduce the numerical equivalent of the continuous $L_2$ loss from Eq. (1) as

$$L = \frac{1}{2} \sum_i \left(\psi_0(\mathcal{D}(\mathbf{x}_i, \boldsymbol{\alpha})) - \phi_{\boldsymbol{\alpha}}(\mathbf{x}_i)\right)^2 \Delta x_i \,, \tag{2}$$

which approximates the spatial integral via the sum over all sample points $i$ with corresponding evaluation positions $\mathbf{x}_i \in \mathrm{R}^4$, where $\Delta x_i = \|\mathbf{x}_i - \mathbf{x}_{i-1}\|$ is constant in our case. This corresponds to a regular grid structure, where the loss is accumulated per cell. The central challenge here is to compute reliable gradients for $\mathcal{D}$, which encapsulates a series of highly non-linear deformation steps. We first focus on inferring $\boldsymbol{\beta}$, with $\mathbf{w} = 0$.

The gradient of Eq. (2) with respect to one component of the deformation parameters $\beta_j$ is then given by

$$\frac{\mathrm{d}}{\mathrm{d}\beta_j} L = \sum_i \left(-\frac{\partial}{\partial \beta_j} \mathbf{v}_{\mathrm{fin}}(\mathbf{x}_i + \mathbf{v}_{\mathrm{inv}}(\mathbf{x}_i, \boldsymbol{\beta}), \boldsymbol{\beta}) \cdot \nabla \psi_0(\mathbf{x}_i - \mathbf{v}_{\mathrm{fin}}(\mathbf{x}_i, \boldsymbol{\beta}))\right) (\psi_0(\mathbf{x}_i, \boldsymbol{\beta}) - \phi_{\boldsymbol{\alpha}}(\mathbf{x}_i)) \,,$$
$$\tag{3}$$

where the first term on the sum over $i$ in parentheses represents the gradient of the deformed initial surface. Here we compute the derivative of the full deformation as $\frac{\partial}{\partial \beta_j} \mathbf{v}_{\mathrm{fin}}(\mathbf{x}_i + \mathbf{v}_{\mathrm{inv}}(\mathbf{x}_i, \boldsymbol{\beta}), \boldsymbol{\beta}) = \mathbf{u}_i^*(\mathbf{x}_i)$. The offset by $\mathbf{v}_{\mathrm{inv}}$ on the left hand side indicates that we perform a forward-advection step for this term. Details of this step, and for the full derivation of the gradient are given in Appendix B.1.

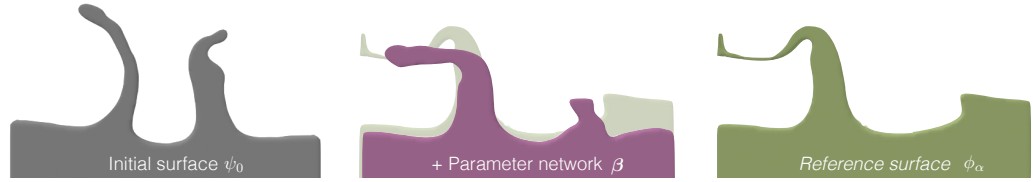

Figure 3: An example of our parameter learning approach. F.l.t.r.: the initial undeformed surface, the surface deformed by the weighting from the trained parameter network, and the reference surface only. The reference surface is shown again in the middle in light brown for comparison. The weighted deformations especially match the left liquid arm well, while there are not enough degrees of freedom in the pre-computed deformations to independently raise the surface on the right side.

A trained NN with this loss functions yields an instance of $f_p$, with which we can infer adjusted deformation weights $f_p(\boldsymbol{\alpha}) = \boldsymbol{\beta}$.

**Learning to Generate Deformations**    Based on $\boldsymbol{\beta}$, we apply the deformation sequence $\mathbf{u}_i$. The goal of our second network, the deformation network $f_d$, is to compute the refinement deformation $\mathbf{w}$. In contrast to the pre-computed $\mathbf{u}_i$, $f_d(\boldsymbol{\alpha}) = \mathbf{w}$ now directly depends on $\boldsymbol{\alpha}$, and can thus capture the interior of the parameter space. Given the initial surface $\psi_0$ deformed by the set of $\beta_i \mathbf{u}_i$, which we will denote as $\tilde{\psi}$ below, the refinement deformation is applied with a final deformation step as $\psi(\mathbf{x}) = \tilde{\psi}(\mathbf{x} - \mathbf{w}(\mathbf{x}, \alpha))$.

In order to compute the gradient of the deformation loss, we introduce the indicator function $\chi_j(\mathbf{x})$ for a single deformation vector $\mathbf{w}_j$ of $\mathbf{w}$. We found it useful to use a fine discretization for the implicit surfaces, such as $\psi$, and lower resolutions for $\mathbf{w}$. Hence, each discrete entry $\mathbf{w}_j$ can act on multiple cells of $\psi$, which we enumerate with the help of $\chi_j$. Now the derivative of Eq. (1) for a fixed $\boldsymbol{\beta}$ with respect to a single deformation vector $\mathbf{w}_j$ of $\mathbf{w}$ is given by

$$\frac{\mathrm{d}}{\mathrm{d}\mathbf{w}_j} L = -\sum_i \chi_j(\mathbf{x}_i) \, \nabla \tilde{\psi}(\mathbf{x}_i - \mathbf{w}(\mathbf{x}_i, \boldsymbol{\alpha})) \, \left( \tilde{\psi}(\mathbf{x}_i, \boldsymbol{\alpha}) - \phi_{\boldsymbol{\alpha}}(\mathbf{x}_i) \right). \tag{4}$$

The full derivation of this gradient is given in Appendix B.2. Our approach for deformation learning can be regarded as an extension of STNs Jaderberg et al. (2015) for dense, weighted fields, and semi-Lagrangian advection methods. The parameter network corresponds to an STN which learns to combine and weight known deformation fields. The deformation network, on the other hand, resembles the thin plate spline STNs, where a network generates an offset for each cell center, which is then used to sample a deformed image or grid. Note that in our case, this sampling process corresponds to the semi-Lagrangian advection of a fluid simulation.

**Training Details**    For $f_p$ we use a simple structure with two fully connected layers, while $f_d$ likewise contains two fully connected layers, followed by two or more four-dimensional de-convolution layers. All layers use ReLU activation functions. Details can be found in App. B, Fig. 12.

In practice, we also found that a small amount of weight decay and $L_2$ regularization of the generated deformations can help to ensure smoothness. Thus, the loss function of the deformation network, with regularization parameters $\gamma_1$ and $\gamma_2$ is

$$L_t = L + \gamma_1 ||\theta||_2 + \gamma_2 ||\mathbf{w}||_2 \, , \tag{5}$$

where $\theta$ denotes the network weights. In addition, regular SDFs can lead to overly large loss values far away from the surface due to linearly increasing distances. Thus, we apply the $\tanh()$ function to the SDF values, in order to put more emphasis on the surface region.

Special care is required for boundary conditions, i.e, the sides of our domain. Assuming constant values outside of the discretized domain, i.e. $\partial\psi(\mathbf{x})/\partial\mathbf{n} = 0$ for all $\mathbf{x}$ at the domain sides leads to vanishing gradients $\nabla\psi(\mathbf{x}) = \mathbf{0}$ in App. B, Eq. (4). We found this causes artificial minima and maxima in the loss function impeding the training process. Hence, we extrapolate the SDF values with $\partial\psi(\mathbf{x})/\partial\mathbf{n} = \pm 1$ in order to retrieve non zero gradients at the domain sides.

To train both networks we use stochastic gradient descent with an ADAM optimizer and a learning rate of $10^{-3}$. Training is performed separately for both networks, with typically 1000 steps for $f_d$,

| | Initial | +Parameters | +Deformation |
|---|---|---|---|
| Liquid 2D | 0.0876 | 0.0521 | 0.0234 |
| Flat | 0.0403 | - | 0.0121 |
| Drop | 0.0431 | 0.024 | 0.0096 |
| Stairs | 0.0953 | 0.0537 | 0.0299 |

Loss (numeric)

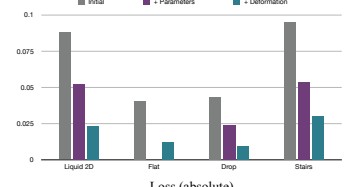

Loss (absolute)

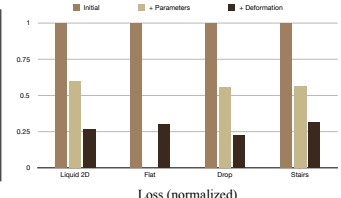

Loss (normalized)

Figure 4: Ablation study for our method. We evaluated the average loss for a test data set of the different data sets discussed in the text. Left: numeric values, again as a graph (center), and a graph of the loss values normalized w.r.t. initial surface loss on the right. Our method achieves very significant and consistent reductions across the very different data sets.

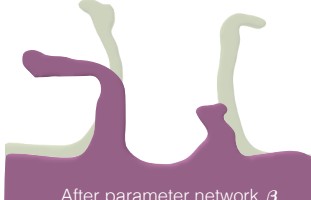
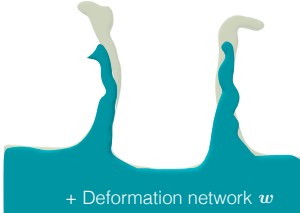
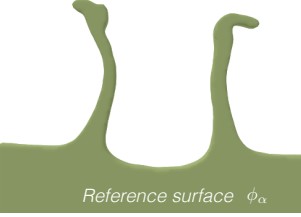

Figure 5: An example of our deformation learning approach. F. l. t. r.: the result after applying weighted deformations, and with an additional deformation from a trained deformation network. Both show the reference surface in light brown in the background, which is shown again for comparison on the right. The inferred deformation manages to reconstruct large parts of the two central arms which can not be recovered by any weighting of the pre-computed deformations (left).

and another ca. 9000 steps for $f_d$. Full parameters can be found in App. B, Table 2. As training data we generate sets of implicit surfaces from liquid simulations with the FLIP method Bridson (2015). For our 2D inputs, we use single time steps, while our 4D data concatenates 3D surfaces over time to assemble a space-time surface. Working in conjunction, our two networks capture highly complex behavior of the fluid space-time surface $\phi_\alpha$ over the whole parameter domain. We will evaluate the influence of the networks in isolation and combination in the following.

## 4 EVALUATION

In order to evaluate our method, we first use a two-dimensional parameter space with two dimensional implicit surfaces from a liquid simulation. An overview of the space of 2156 training samples of size $100^2$ can be found in the supplemental materials. For our training and synthesis runs, we typically downsample the SDF data, and use a correspondingly smaller resolution for the output of the deformation network, see Appendix C.2, Table 2. The effect of our trained networks in terms of loss reductions is shown on the left side of Fig. 4 under *Liquid 2D*. As baseline we show the loss for the undeformed surface w.r.t. the test data samples. For this 2D data set, employing the trained parameter network reduces the loss to $59.4\%$ of the initial value. Fig. 3 shows the surface of an exemplary result. Although our result does not exactly match the target due to the constraints of the pre-computed deformations, the learned deformation weights lead to a clear improvement in terms of approximating the target.

The inferred deformation of our deformation network further reduces the surface loss to $26.6\%$ of its initial value, as shown in Fig. 4. This is equivalent to a reduction to $44.8\%$ compared the result after applying the weighted deformations. Note that the graphs in this figure correspond to an ablation study: starting with the loss for the undeformed surface, over using only the parameter network, to deformations for a flat surface, to our full algorithm. An example surface for these two-dimensional cases can be found in Fig. 5. This figure compares the surface after applying weighted and inferred deformations, i.e. our full method (right), with a surface deformed by only by deformations weighted by the parameter network (left). The NN deformation manages to reconstruct the two arm in the center of the surface, which the pre-computed deformations fail to capture. It is also apparent

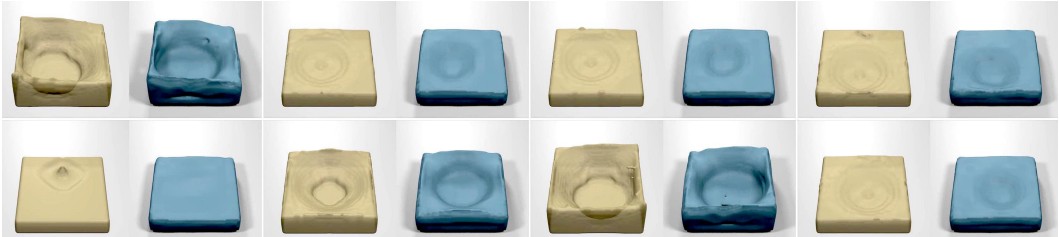

Figure 6: Eight examples of the learned deformations for a flat initial surface. For each pair the reference surfaces are depicted in yellow and the deformed results in blue. The trained model learns to recover a significant portion of the large-scale surface motion over the whole parameters space.

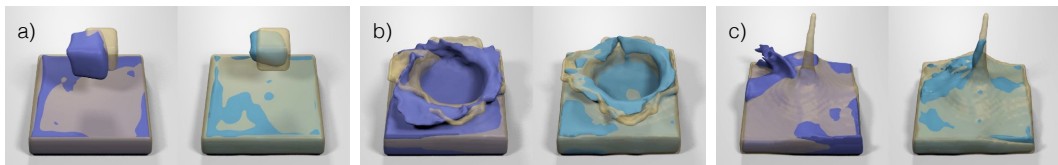

Figure 7: Each pair shows the reference surface in transparent brown, and in purple on the left the deformed surface after applying the precomputed deformations. These surfaces often significantly deviate from the brown target, i.e. the visible purple regions indicates misalignments. In cyan on the right, our final surfaces based on the inferred deformation field. These deformed surface match the target surface closely, and even recover thin features such as the central peak in (c).

that despite the improvement, this surface does not reach the tip of the arms. This is caused by regularization over the varying set of target surfaces, leading to an averaged solution for the deformation. Additional examples for this two dimensional setup can be found in the supplemental video.

**4D Surface Data**   Next we consider complete space-time data sets in four dimensions. with a three dimensional parameter space $\boldsymbol{\alpha}$. The three parameter dimensions are $x$- and $y$-coordinates of the initial drop position, as well as its size. We use a total of 1764 reference SDFs with an initial resolution of $100^4$, which are down-sampled to a resolution of $40^4$. To illustrate the capabilities of the deformation network, we start with a completely flat initial surface as $\psi_0$, and train the deformation network to recover the targets. As no pre-computed deformations are used for this case, we do not train a parameter network. The flat initial surface represents an especially tough case, as the network can not rely on any small scale details in the reference to match with the features of the targets. Despite this difficulty, the surface loss is reduced to $30\%$ of the initial loss purely based on the deformation network. A set of visual examples can be seen in Fig. 6. Due to the reduced resolution of the inferred deformation w.r.t. the SDF surface, not all small scale features of the targets are matched. However, the NN manages to reconstruct impact location and size very well across the full parameter space. Additional 2D and 4D results can be found in the supplemental materials.

Once we introduce a regular initial surface for $\psi_0$, in this case the zero parameter configuration with the drop in one corner of the domain with the largest size, our networks perform even better than for the 2D data discussed above. The weighted deformations lead to a loss reduction to $55.6\%$ of the initial value, and the learned deformation reduce the loss further to $22.2\%$ of the baseline loss (Fig. 4). An example is shown in Fig. 7. In contrast to the flat surface test, the network deformation can now shift and warp parts of $\psi_0$, such as the rim of the splash of Fig. 7 to match the targets.

## 5   ADDITIONAL RESULTS WITH IMPLICIT SURFACES IN 4D

Our method yields highly reduced representations which can be used to very efficiently synthesize simulation results. To demonstrate the representational capabilities and the performance of our method, we have integrated the evaluation pipeline for our trained networks into an Android application. As our method yields an implicit surface as 4D array, visualizing the resulting animation is very efficient. We render slices of 3D data as implicit surfaces, and the availability of a full

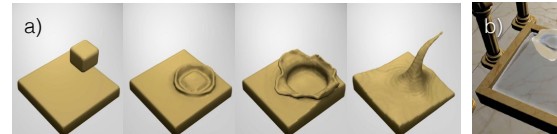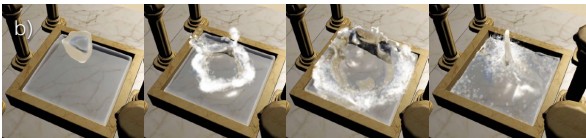

Figure 8: a) Liquid drop data set example: several 3D surfaces of a single simulation data point in $\phi_{\boldsymbol{\alpha}}$. b) An example splash generated by our method, visualized interactively.

3D representations makes it possible to add curvature-based shading and secondary particle effects on the fly. In this context, please also consider the supplemental materials, which contain these sequences in motion. They are available at: `https://ge.in.tum.de/publications/2017-prantl-defonn/`.

**Performance**  One of the setup available in this app is the liquid drop setup with 3D parameter space described above. With this setup a user can release drops of liquid at varying positions and of varying size. An example reference and generated result can be found in Fig. 8. For this liquid drop setup, evaluating the network takes 69ms on average, and assembling the final deformation field another 21.5ms. We use double buffering, and hence both tasks are evaluated in a background thread. Rendering the implicit surface takes an average of 21ms, with an average frame rate of 50 fps. The original simulation for the drop setup of Fig. 8 took 530 seconds on average with a parallel implementation to generate a single 4D surface data point.Assuming a best-case slowdown of only 4x for the mobile device, it would require more than 32 minutes to run the original simulation there. Our app generates and renders a full liquid animation in less than one second in total. Thus, our algorithm generates the result roughly 2000 times faster than the regular simulation. Our approach also represents the space of more than 1700 input simulations, i.e., more than 17GB, with less than 30MB of storage.

**Stairs**  A next setup, shown in Fig. 9, captures a continuous flow around a set of obstacles. Liquid is generated in one corner of the simulation domain, and then flows in a U-shaped path around a wall, down several steps. In the interactive visualization, green arrows highlight in- and outflow regions. The three dimensional parametrization of this setup captures a range of positions for the wall and two steps leading to very different flow behaviors for the liquid. In this case the data set consists of 1331 SDFs, and our app uses an output resolution of $50^4$. The corresponding loss measurements can be found in the right graphs of Fig. 4. As with the two previously discussed data sets, our approach leads to very significant reductions of the surface loss across the full parameter space, with a final residual loss of $31.3\%$ after applying the learned deformation. Due to larger size of the implicit surfaces and the inferred deformation field, the performance reduces to a frame rate of 30 fps on average, which, however, still allows for responsive user interactions.

**Discussion**  Our approach in its current form has several limitations that are worth mentioning. E.g., we assume that the space of target surfaces have a certain degree of similarity, such that a single surface can be selected as initial surface $\psi_0$. In addition, our method currently does not make use of the fact that the inputs are generated by a physical process. E.g., it would be highly interesting for future work to incorporate additional constraints such as conservation laws, as currently our results can deviate from an exact conservation of physical properties. E.g., due to its approximating nature our method can lead to parts of the volume disappearing and appearing over time. Additionally, the L2 based loss can lead to rather smooth results, here approaches such as GANs could potentially improve the results.

## 6  CONCLUSIONS

We have presented a novel method to generate space-time surfaces with deformation-aware neural networks. In particular, we have demonstrated the successful inference of weighting sequences of aligned deformations, and the generation of dense deformation fields across a range of varied inputs. Our method exhibits significant improvements in terms surface reconstruction accuracy across the full parameter range. In this way, our networks can capture spaces of complex surface behavior, and

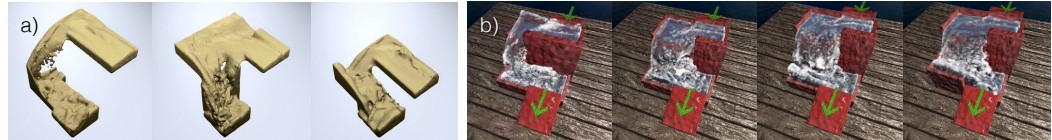

Figure 9: a) Three example configurations from our stairs data set. b) The interactive version of the stair setup shown in the demo app. Notice how the flow around the central wall obstacle changes. As the wall is shifted right, the flow increases corresonpondingly.

allow for real-time interactions with physics effects that are orders of magnitudes slower to compute with traditional solvers.

Beyond liquid surfaces, our deformation networks could also find application for other types of surface data, such as those from object collections or potentially also moving characters. Likewise, it could be interesting to extend our method in order to infer deformations for input sets without an existing parametrization.

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

# Appendix: Generating Liquid Simulations with Deformation-aware Neural Networks

This supplemental document will first detail the necessary steps to align multiple, weighted deformation fields. Afterwards, we will derive the gradients presented in the paper for the parameter and deformation networks, and then present additional results.

## A    DEFORMATION ALIGNMENT

As before, $\phi_{\boldsymbol{\alpha}}$ denotes the reference signed distance functions (SDFs) of our input parameter space, while $\psi$ denotes instances of a single input surface, typically deformed by our algorithm. We will denote the single initial surface without any deformations applied with $\psi_0$. Here, we typically use the zero point of our parameter space, i.e., $\psi_0 = \phi_{\alpha_0}$, with $\alpha_0 = \mathbf{0}$. Hence, we aim for deforming $\psi_0$ such that it matches all instances of $\phi_{\boldsymbol{\alpha}}$ as closely as possible.

For the pre-computed, end-point deformations, it is our goal to only use a single deformation for each dimension of the parameter space $\boldsymbol{\alpha}$. Thus $\mathbf{u}_1$ will correspond to $\alpha_1$ and be weighted by $\beta_1$, and we can apply $\beta_1 \mathbf{u}_1$ to compute a deformation for an intermediate point along this dimension. Given the sequence of pre-computed deformations $\{\mathbf{u}_1, \mathbf{u}_2, \ldots, \mathbf{u}_N\}$ and a point in parameter space $\{\beta_1, \ldots, \beta_N\}$ a straight-forward approach is to apply each deformation sequentially

$$\psi_1(\mathbf{x}, \boldsymbol{\beta}) = \psi_0(\mathbf{x} - \beta_1 \mathbf{u}_1)$$
$$\psi_2(\mathbf{x}, \boldsymbol{\beta}) = \psi_1(\mathbf{x} - \beta_2 \mathbf{u}_2)$$
$$\vdots$$
$$\psi_N(\mathbf{x}, \boldsymbol{\beta}) = \psi_{N-1}(\mathbf{x} - \beta_N \mathbf{u}_N). \tag{6}$$

However, there are two disadvantages to this approach. The main problem is that the deformations $\mathbf{u}_i$ are only meaningful if applied with $\beta_i = 1$.

Thus, if a previous deformation wasn't applied fully with a weight of 1, each subsequent deformation will lead to an accumulation of deformation errors. The second disadvantage of this simple method

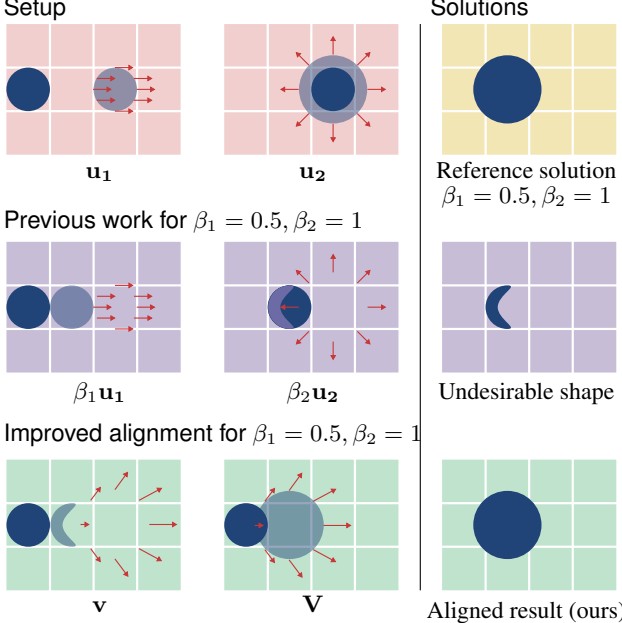

Figure 10: Illustration of our deformation alignment procedure.

is the fact that many advection steps have to be applied in order to arrive at the final deformed SDF. This also affects performance as each advection step introduces additional computations, and scattered memory accesses. This is best illustrated with the an example, shown in Fig. 10. In row 1) the first two figures in red show two pre-computed end-point deformations (red arrows). The first one ($\mathbf{u}_1$) moves a drop to the right, while $\mathbf{u}_2$ changes its size once it has reached its final position. Images with deformations show the source in dark blue, and the deformed surface in light blue. In this example, the two deformations should be combined such that the horizontal position and the drop size can be independently controlled by changing $\beta_1$ and $\beta_2$. E.g., on the top right, a correct solution for $\beta_1 = 0.5, \beta_2 = 1$ is shown. Row 2) of Fig. 10 shows how these deformations are applied in previous work: The second deformation acts on wrong parts of the surface, as the drop has not reached its left-most position for $\beta_1 = 0.5$. The undesirable result in this case is a partially deformed drop, shown again in the middle row on the right.

We present an alternative approach which aligns the deformation fields to the final position of the deformation sequence. Then, all aligned deformation fields can simply be accumulated by addition, and applied to the input in a single step. To do this, we introduce the intermediate deformation fields:

$$\mathbf{u}_N^*(\mathbf{x}) = \mathbf{u}_N(\mathbf{x}),$$
$$\mathbf{u}_{N-1}^*(\mathbf{x}) = \mathbf{u}_{N-1}(\mathbf{x} - \mathbf{u}_N^*(\mathbf{x})),$$
$$\mathbf{u}_{N-2}^*(\mathbf{x}) = \mathbf{u}_{N-2}(\mathbf{x} - \mathbf{u}_N^*(\mathbf{x}) - \mathbf{u}_{N-1}^*(\mathbf{x})),$$
$$\vdots$$
$$\mathbf{u}_1^*(\mathbf{x}) = \mathbf{u}_1(\mathbf{x} - \mathbf{u}_N^*(\mathbf{x}) - \mathbf{u}_{N-1}^*(\mathbf{x}) \ldots - \mathbf{u}_2^*(\mathbf{x})). \tag{7}$$

Each $\mathbf{u}_i^*$ is moved by all subsequent deformations $\mathbf{u}_j^*, j \in [i+1 \cdots N]$, such that it acts on the correct target position under the assumption that $\beta_i = 1$ (we will address the case of $\beta_i \neq 1$ below). The Eulerian representation we are using means that advection steps look backward to gather data, which in our context means that we start with the last deformation $\mathbf{u}_N$ to align previous deformations. Using the aligned deformation fields $\mathbf{u}^*$ we can include $\boldsymbol{\beta}$ and assemble the weighted intermediate fields

$$\mathbf{v}_{\text{sum}}(\mathbf{x}, \boldsymbol{\beta}) = \sum_{i=1}^{N} \beta_i \mathbf{u}_i^*(\mathbf{x}) \tag{8}$$

and an inversely weighted correction field

$$\mathbf{v}_{\text{inv}}(\mathbf{x}, \boldsymbol{\beta}) = -\sum_{i=1}^{N} (1 - \beta_i) \mathbf{u}_i^*(\mathbf{x}). \tag{9}$$

The first deformation field $\mathbf{v}_{\text{sum}}$ represents the weighted sum of all aligned deformations, weighted with the correct amount of deformation specified by the deformation weights $\beta_i$. The second deformation $\mathbf{v}_{\text{inv}}$ intuitively represents the offset of the deformation field $\mathbf{v}_{\text{sum}}$ from its destination caused by the $\boldsymbol{\beta}$ weights. Therefore, we correct the position of $\mathbf{v}_{\text{sum}}$ by this offset with the help of an additional forward-advection step calculated as:

$$\mathbf{v}_{\text{fin}}(\mathbf{x} + \mathbf{v}_{\text{inv}}(\mathbf{x}, \boldsymbol{\beta}), \boldsymbol{\beta}) = \mathbf{v}_{\text{sum}}(\mathbf{x}, \boldsymbol{\beta}), \tag{10}$$

This gives us the final deformation field $\mathbf{v}_{\text{fin}}(\mathbf{x}, \boldsymbol{\beta})$. It is important to note that the deformation $\mathbf{v}_{\text{sum}}$ for a position $\mathbf{x}$ is assembled at a location $\mathbf{x}'$ that is not known a-priori. It has to be transported to $\mathbf{x}$ with the help of $\mathbf{v}_{\text{inv}}$, as illustrated in Fig. 11.

This correction is not a regular advection step, as the deformation is being 'pushed' from $\mathbf{x} + \mathbf{v}_{\text{inv}}(\mathbf{x}, \boldsymbol{\beta})$ to $\mathbf{x}$. In order to solve this advection equation we use an inverse semi-Lagrangian step, inspired by algorithms such as the one by Lentine et al. Lentine et al. (2011), pushing values forward with linear interpolation. As multiple values can end up in a single location, we normalize their contribution. Afterwards, we perform several iterations of a "fill-in" step to make sure all cells in the target deformation grid receive a contribution (we simply extend and average deformation values from all initialized cells into uninitialized regions).

The deformed SDF is then calculated with a regular advection step applying the final, aligned deformation with

$$\psi(\mathbf{x}, \boldsymbol{\beta}) = \psi_0(\mathbf{x} - \mathbf{v}_{\text{fin}}(\mathbf{x}, \boldsymbol{\beta})). \tag{11}$$

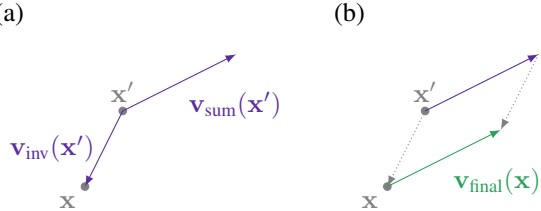

Figure 11: This figure illustrates the forward advection process: Both deformation $\mathbf{v}_{\text{sum}}$ and the correction $\mathbf{v}_{\text{inv}}$ are initially located at $\mathbf{x}'$ in (a). $\mathbf{v}_{\text{inv}}$ is applied to yield the correct deformation at location $\mathbf{x}$, as shown in (b).

Based on our correction step from Eq. (10) this method now respects the case of partially applied deformations. As the deformations $\mathbf{u}^*(\mathbf{x})$ are already aligned and don't depend on $\boldsymbol{\beta}$, we can pre-compute them. To retrieve the final result it is now sufficient to sum up all deformations in $\mathbf{v}_{\text{sum}}$ and $\mathbf{v}_{\text{inv}}$, then apply one forward-advection step to compute $\mathbf{v}_{\text{fin}}$, and finally deform the input SDF by applying semi-Lagrangian advection. While our method is identical with alignment from previous work Thuerey (2017) for $\beta_i = 1$, it is important for practical deformations with weights $\beta_i \neq 1$.

Our method is illustrated in row 3) of Fig. 10. In the bottom left we see the deformation field $\mathbf{v}_{\text{sum}}$ from previous work. It is also the starting point of our improved alignment, but never applied directly. Our method corrects $\mathbf{v}_{\text{sum}}$ by transforming it into $\mathbf{v}_{\text{fin}}$, bottom center, which acts on the correct spatial locations. In this example, it means that the expanding velocities from $\mathbf{u}_2$ are shifted left to correctly expand the drop based on its initial position. Our method successfully computes the intended result, as shown in the bottom right image.

This algorithm for aligning deformations will be our starting point for learning the weights $\boldsymbol{\beta}$. After applying the weighted deformations, we adjust the resulting surface we an additional deformation field generated by a trained model. In the following, we will derive gradients for learning the weighting as well as the refinement deformation.

## B  LEARNING DEFORMATIONS

As outlined in the main document, we aim for minimizing the $L_2$ loss between the final deformed surface and the set of reference surfaces $\phi_{\boldsymbol{\alpha}}$, i.e.:

$$L = \frac{1}{2} \sum_i \left( \psi_0(\mathcal{D}(\mathbf{x}_i, \boldsymbol{\alpha})) - \phi_{\boldsymbol{\alpha}}(\mathbf{x}_i) \right)^2 , \tag{12}$$

where $\mathcal{D}(\mathbf{x}_i, \boldsymbol{\alpha})$ denotes the joint application of all weighted and generated deformation fields.

### B.1  LEARNING DEFORMATION WEIGHTING

We will first focus on the parameter network to infer the weighting of the pre-computed deformation fields based on the input parameters $\boldsymbol{\alpha}$. Thus, the NN has the goal to compute $\boldsymbol{\beta}(\boldsymbol{\alpha}) \in \mathrm{R}^N = (\beta_1(\boldsymbol{\alpha}), \ldots, \beta_N(\boldsymbol{\alpha}))$ in order to minimize Eq. (12). The application of the deformations weighted by $\boldsymbol{\beta}$ includes our alignment step from Sec. A, and hence the neural networks needs to be aware of its influence. To train the parameter network, we need to specify gradients of Eq. (12) with respect to the network weights $\theta_i$. With the chain rule we obtain $\frac{\mathrm{d}}{\mathrm{d}\theta_{ij}^l} L = \frac{\mathrm{d}\boldsymbol{\beta}}{\mathrm{d}\theta_{ij}^l} \frac{\mathrm{d}L}{\mathrm{d}\boldsymbol{\beta}}$. Since the derivative of the network output $\beta_i$ with respect to a specific network weight $\theta_{ij}^l$ is easily calculated with backpropagation Bishop (2006), it is sufficient for us to specify the second term. The gradient of Eq. (12) with respect to the deformation parameter $\beta_i$ is given by

$$\frac{\mathrm{d}}{\mathrm{d}\beta_i} L = \sum_j \frac{\mathrm{d}}{\mathrm{d}\beta_i} \psi(\mathbf{x}_j, \boldsymbol{\beta}) \left[ \psi(\mathbf{x}_j, \boldsymbol{\beta}) - \phi_{\boldsymbol{\alpha}}(\mathbf{x}_j) \right], \tag{13}$$

where we have inserted Eq. (11). While the second term in the sum is easily computed, we need to calculate the first term by differentiating Eq. (11) with respect to $\beta_i$, which yields

$$\frac{\mathrm{d}}{\mathrm{d}\beta_i}\psi(\mathbf{x},\boldsymbol{\beta}) = -\frac{\mathrm{d}}{\mathrm{d}\beta_i}\mathbf{v}_{\mathrm{fin}}(\mathbf{x},\boldsymbol{\beta}) \cdot \nabla\psi_0(\mathbf{x} - \mathbf{v}_{\mathrm{fin}}(\mathbf{x},\boldsymbol{\beta})). \tag{14}$$

As the gradient of $\psi_0$ is straight forward to compute, $\frac{\mathrm{d}}{\mathrm{d}\beta_i}\mathbf{v}_{\mathrm{fin}}(\mathbf{x},\boldsymbol{\beta})$ is crucial in order to compute a reliable derivative. It is important to note that even for the case of small corrections $\mathbf{v}_{\mathrm{inv}}(\mathbf{x},\boldsymbol{\beta})$, Eq. (10) cannot be handled as another backward-advection step such as $\mathbf{v}_{\mathrm{fin}}(\mathbf{x},\boldsymbol{\beta}) = \mathbf{v}_{\mathrm{sum}}(\mathbf{x} - \mathbf{v}_{\mathrm{inv}}(\mathbf{x},\boldsymbol{\beta}),\boldsymbol{\beta})$. While it might be tempting to assume that differentiating this advection equation will produce reasonable outcomes, it can lead to noticeable errors in the gradient. These in turn quickly lead to diverging results in the learning process, due to the non-linearity of the problem.

The correct way of deriving the change in $\mathbf{v}_{\mathrm{fin}}(\mathbf{x},\boldsymbol{\beta})$ is by taking the total derivative of $\mathbf{v}_{\mathrm{sum}}(\mathbf{x},\boldsymbol{\beta}) = \mathbf{v}_{\mathrm{fin}}(\mathbf{x} + \mathbf{v}_{\mathrm{inv}}(\mathbf{x},\boldsymbol{\beta}),\boldsymbol{\beta})$ with respect to $\beta_i$:

$$\frac{\mathrm{d}}{\mathrm{d}\beta_i}\mathbf{v}_{\mathrm{sum}}(\mathbf{x},\boldsymbol{\beta})$$
$$= \frac{\partial}{\partial\beta_i}\mathbf{v}_{\mathrm{fin}}(\mathbf{x} + \mathbf{v}_{\mathrm{inv}}(\mathbf{x},\boldsymbol{\beta}),\boldsymbol{\beta}) + \mathbf{J}_V(\mathbf{x} + \mathbf{v}_{\mathrm{inv}}(\mathbf{x},\boldsymbol{\beta}),\boldsymbol{\beta})\frac{\partial}{\partial\beta_i}\mathbf{v}_{\mathrm{inv}}(\mathbf{x},\boldsymbol{\beta}), \tag{15}$$

where, $\mathbf{J}_V(\mathbf{x} + \mathbf{v}_{\mathrm{inv}}(\mathbf{x},\boldsymbol{\beta}),\boldsymbol{\beta})$ denotes the Jacobian of $\mathbf{v}_{\mathrm{fin}}$ with respect to $\mathbf{x}$, evaluated at $\mathbf{x} + \mathbf{v}_{\mathrm{inv}}(\mathbf{x},\boldsymbol{\beta})$. Rearranging Eq. (15) and inserting $\mathbf{v}_{\mathrm{sum}}$ and $\mathbf{v}_{\mathrm{inv}}$ yields

$$\frac{\partial}{\partial\beta_i}\mathbf{v}_{\mathrm{fin}}(\mathbf{x} + \mathbf{v}_{\mathrm{inv}}(\mathbf{x},\boldsymbol{\beta}),\boldsymbol{\beta}) \tag{16}$$
$$= \frac{\mathrm{d}}{\mathrm{d}\beta_i}\mathbf{v}_{\mathrm{sum}}(\mathbf{x},\boldsymbol{\beta}) - \mathbf{J}_V(\mathbf{x} + \mathbf{v}_{\mathrm{inv}}(\mathbf{x},\boldsymbol{\beta}),\boldsymbol{\beta})\frac{\partial}{\partial\beta_i}\mathbf{v}_{\mathrm{inv}}(\mathbf{x},\boldsymbol{\beta})$$
$$= \frac{\mathrm{d}}{\mathrm{d}\beta_i}\sum_{i=1}^{N}\beta_i\mathbf{u}_i^*(\mathbf{x}) + \mathbf{J}_V(\mathbf{x} + \mathbf{v}_{\mathrm{inv}}(\mathbf{x},\boldsymbol{\beta}),\boldsymbol{\beta})\frac{\partial}{\partial\beta_i}\sum_{i=1}^{N}(1 - \beta_i)\mathbf{u}_i^*(\mathbf{x})$$
$$= [\mathbf{1} - \mathbf{J}_V(\mathbf{x} + \mathbf{v}_{\mathrm{inv}}(\mathbf{x},\boldsymbol{\beta}),\boldsymbol{\beta})]\mathbf{u}_i^*(\mathbf{x}). \tag{17}$$

We note that the Jacobian in the equation above has small entries due to the smooth nature of the deformations $\mathbf{v}_{\mathrm{fin}}$. Thus, compared to the unit matrix it is small in magnitude. Note that this relationship is not yet visible in Eq. (15). We have verified in experiments that $\mathbf{J}_V$ does not improve the gradient significantly, and we thus set this Jacobian to zero, arriving at

$$\frac{\partial}{\partial\beta_i}\mathbf{v}_{\mathrm{fin}}(\mathbf{x} + \mathbf{v}_{\mathrm{inv}}(\mathbf{x},\boldsymbol{\beta}),\boldsymbol{\beta}) \approx \mathbf{u}_i^*(\mathbf{x}), \tag{18}$$

where the $\mathbf{u}^*$ are the deformation fields aligned for the target configuration from Eq. (7). We use Eq. (18) to estimate the change in the final deformation fields for changes of the $i$-th deformation parameter. We see that this equation has the same structure as Eq. (10). On the left-hand side, we have $\frac{\partial}{\partial\beta_i}\mathbf{v}_{\mathrm{fin}}$, evaluated at $\mathbf{x} + \mathbf{v}_{\mathrm{inv}}(\mathbf{x},\boldsymbol{\beta})$, whereas $\mathbf{u}_i^*$ on the right-hand side is evaluated at $\mathbf{x}$. To calculate $\frac{\mathrm{d}}{\mathrm{d}\beta_i}\mathbf{v}_{\mathrm{fin}}(\mathbf{x},\boldsymbol{\beta})$ then, we can use the same forward-advection algorithm, which is applied to the correction in Eq. (10). With this, we have all the necessary components to assemble the gradient from Eq. (13) for training the parameter network with back-propagation.

## B.2   LEARNING TO GENERATE DEFORMATIONS

Our efforts so far have been centered around producing a good approximation of $\phi_{\boldsymbol{\alpha}}$, with a set of given end-point deformations $\{\mathbf{u}_0, \ldots, \mathbf{u}_n\}$. The performance of this method is therefore inherently constrained by the amount of variation we can produce with the deformation inputs. To allow for more variation, we propose to generate an additional space-time deformation field $\mathbf{w}(\boldsymbol{\alpha})$, that changes with the simulation parameters $\boldsymbol{\alpha}$. Once again, we model this function with a neural network, effectively giving the network more expressive capabilities to directly influence the final deformed surface.

For this network we choose a structure with a set of four-dimensional deconvolution layers that generate a dense space-time deformation field. We apply the trained deformation with an additional

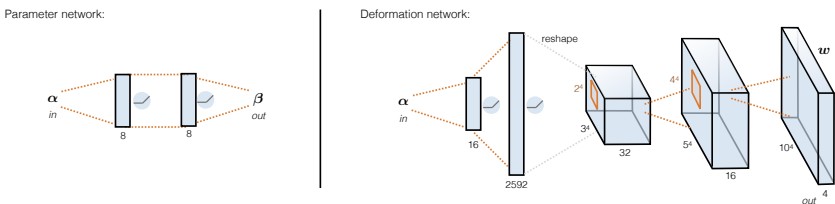

Figure 12: Overview of our two neural networks. While the parameter network (left) is simple, consisting of two fully connected layers, its cost functions allows it to learn how to apply multiple long-range, non-linear deformation fields. The deformation network (right), which makes use of several de-convolutional layers, instead learns to generate dense deformation fields to refine the final surface.

advection step after applying the deformations weighted by the parameter network:

$$\tilde{\psi}(\mathbf{x}) = \psi_0 \left( \mathbf{x} - \mathbf{v}_{\text{fin}}(\mathbf{x}, \boldsymbol{\beta}(\boldsymbol{\alpha})) \right), \tag{19}$$

$$\psi(\mathbf{x}) = \tilde{\psi} \left( \mathbf{x} - \mathbf{w}(\mathbf{x}, \boldsymbol{\alpha}) \right). \tag{20}$$

Thus, the deformation network only has to learn to refine the surface $\tilde{\psi}$ after applying the weighted deformations, in order to accommodate the nonlinear behavior of $\phi_{\boldsymbol{\alpha}}$.

As input, we supply the deformation network with the simulation parameters $\boldsymbol{\alpha} = (\alpha_i, \dots, \alpha_N)$ as a vector. The output of the network are four-component vectors, with the resolution $R_x \times R_y \times R_z \times R_t$. Note that in general the SDF resolution and the deformation resolution do not need to be identical. Given a fixed SDF resolution, we can use a smaller resolution for the deformation, which reduces the number of weights and computations required for training. Thus in practice, each four-dimensional vector of the deformation acts on a region of the SDF, for which we assume the deformation to be constant. Therefore, we write the deformation field as

$$\mathbf{w}(\mathbf{x}, \boldsymbol{\alpha}) = \sum_j \xi_j(\mathbf{x}) \, \mathbf{w}_j(\boldsymbol{\alpha}), \tag{21}$$

where $\xi_j(\mathbf{x})$ is the indicator function of the $j$-th region on which the four-dimensional deformation vector $\mathbf{w}_j(\boldsymbol{\alpha})$ acts. This vector is the $j$-th output of the deformation network.

For training, we need to calculate the gradient of the loss-function Eq. (12) with respect to the network weights. Just like in the previous section, it is sufficient to specify the gradient with respect to the network outputs $\mathbf{w}_i(\boldsymbol{\alpha})$. Deriving Eq. (12) yields

$$\frac{\mathrm{d}}{\mathrm{d}\mathbf{w}_i} L$$

$$= \sum_j \frac{\mathrm{d}}{\mathrm{d}\mathbf{w}_i} \psi(\mathbf{x}) \left( \psi(\mathbf{x}) - \phi_{\boldsymbol{\alpha}}(\mathbf{x}) \right)$$

$$= \sum_j \frac{\mathrm{d}}{\mathrm{d}\mathbf{w}_i} \tilde{\psi} \left( \mathbf{x} - \mathbf{w}(\mathbf{x}, \boldsymbol{\alpha}) \right) \; \left( \psi(\mathbf{x}) - \phi_{\boldsymbol{\alpha}}(\mathbf{x}) \right)$$

$$= - \sum_j X_i(\mathbf{x}_j) \, \nabla \tilde{\psi}(\mathbf{x}_j - \mathbf{w}(\mathbf{x}_j, \boldsymbol{\alpha})) \; \left( \psi(\mathbf{x}_j, \boldsymbol{\alpha}) - \phi_{\boldsymbol{\alpha}}(\mathbf{x}_j) \right). \tag{22}$$

Thus, we can calculate the derivative by summation over the region that is affected by the network output $\mathbf{w}_i$. The gradient term is first calculated by evaluating a finite difference stencil on $\tilde{\psi}(\mathbf{x}_j)$ and then advecting it with the corresponding deformation vector $\mathbf{w}(\mathbf{x}_j, \boldsymbol{\alpha})$. The other terms in Eq. (22) are readily available. Alg. 1 summarizes our algorithm for training the deformation network. In particular, it is important to deform the input SDF gradients with the inferred deformation field, in order to calculate the loss gradients in the correct spatial location for backpropagation.

---

**ALGORITHM 1:** Training the deformation network

---

**Data:** training samples from $\phi_{\boldsymbol{\alpha}}$
**Result:** trained deformation network weights $\Theta$
**for** *each training sample* $\{\tilde{\boldsymbol{\alpha}}, \tilde{\phi}\}$ **do**
    evaluate neural network to compute $\beta(\tilde{\boldsymbol{\alpha}})$
    load reference SDF $\tilde{\phi}$, initial SDF $\psi_0$
    calculate $\mathbf{v}_{\text{fin}}(\mathbf{x}_i, \beta(\tilde{\boldsymbol{\alpha}}))$
    $\tilde{\psi}$ = advect $\psi_0$ with $\mathbf{v}_{\text{fin}}$
    calculate $\nabla\tilde{\psi}$
    evaluate neural network to compute $\mathbf{w}_i(\tilde{\boldsymbol{\alpha}})$
    assemble $\mathbf{w}(\mathbf{x}_i)$ from $\mathbf{w}_i(\tilde{\boldsymbol{\alpha}}, \Theta)$ according to Eq. (21)
    advect $\tilde{\psi}$ with $\mathbf{w}$
    advect $\nabla\tilde{\psi}$ with $\mathbf{w}$
    **for** *each* $\mathbf{w}_i$ **do**
        calculate the gradient $\frac{\mathrm{d}}{\mathrm{d}\mathbf{w}_i}L$ according to Eq. (22)
    **end**
    backpropagate $\frac{\mathrm{d}}{\mathrm{d}\mathbf{w}_i}L_t$ from Eq. (5) to adjust $\Theta$
**end**

---

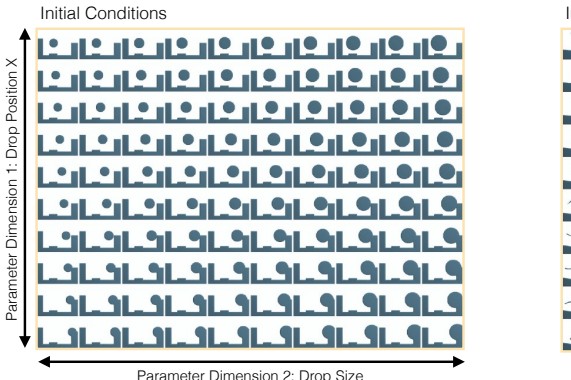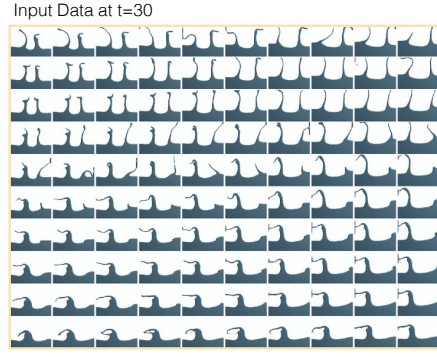

Figure 13: The left image illustrates the initial conditions of our two dimensional parameter space setup. It consists of a set of two-dimensional liquid simulations, which vary the position of the liquid drop along x as $\alpha_1$, and its size as $\alpha_2$. The right half shows the data used for training at $t = 30$. Note the significant amount of variance in positions of small scale features such as the thin sheets. Both images show only a subset of the whole data.

## C  ADDITIONAL EVALUATION

### C.1  2D DATA SET

In the following, we explain additional details of the evaluation examples. For the two dimensional data set, we use the SDFs extracted from 2D simulations of a drop falling into a basin. As simulation parameters we choose $\alpha_1$ to be the size of the drop, and $\alpha_2$ to be its initial $x$-position, as shown in Fig. 13. From this simulation we extract a single frame at $t = 30$, which gives us a two-dimensional parameter-space $\boldsymbol{\alpha} = (\alpha_1, \alpha_2)$, where each instance of $\boldsymbol{\alpha}$ has a corresponding two-dimensional SDF. In order to train the networks described in section 3, we sample the parameter domain with a regular $44 \times 49$ grid, which gives us 2156 training samples, of which we used 100 as a validation set.

Fig. 14 shows the validation loss and the training loss over the iterations both for parameter learning and for deformation learning. We observe that in both cases the learning process reduces the loss, and finally converges to a stable solution. This value is lower in the case of deformation training, which can be easily explained with the increased expressive capabilities of the deformation network.

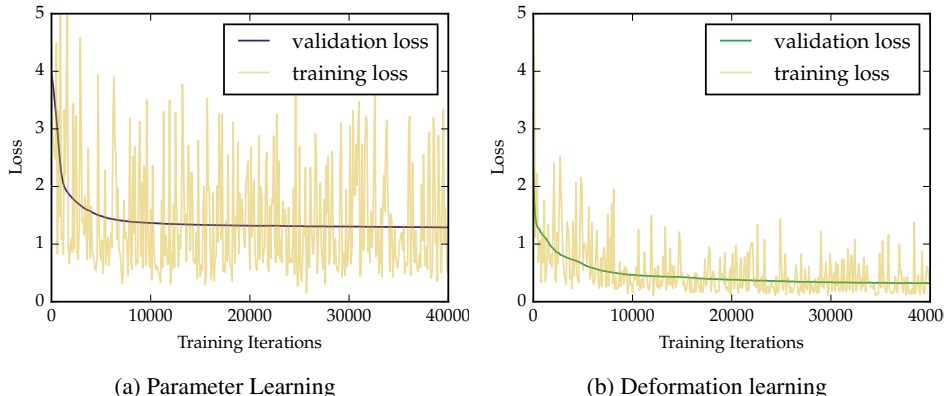

(a) Parameter Learning            (b) Deformation learning

Figure 14: Loss during training both for parameter learning and deformation learning. In yellow we show the loss for the current sample, while the dark line displays the loss evaluated on the validation set.

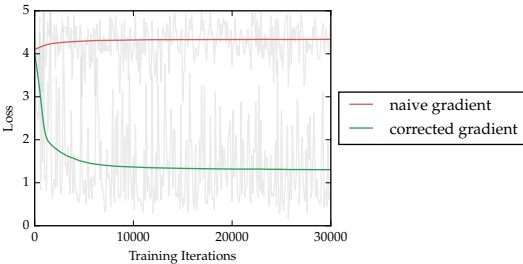

Figure 15: Training with different gradient approximations: validation loss with a simplified advection (red), and the correct gradient from forward advection (green). The simplified version does not converge.

We verified that the solution converged by continuing training for another 36000 steps, during which the change of the solution was negligible.

As mentioned above, it might seem attractive to use a simpler approximation for the forward advection in Eq. (13), i.e., using a simpler, regular advection step. However, due to the strong non-linearity of our setting, this prevents the network from converging, as shown in Fig. 15.

The effect of our deformation network approach is illustrated in Fig. 5. This figure compares our full method (on the right) with several other algorithms. A different, but popular approach for non-linear dimensionality reduction, which can be considered as an alternative to our method, is to construct a reduced basis with PCA. Using the mean surface with four eigenvectors yields a similar reduction to our method in terms of memory footprint. We additionally re-project the different reference surfaces into the reduced basis to improve the reconstruction quality of the PCA version. However, despite this the result is a very smooth surface that fails to capture any details of the behavior of the parameter space, as can be seen in the left column of Fig. 5.

The next column of this figure (in pink) shows the surfaces obtained with the learned deformation weights with our parameter network (Fig. 12 top), but without an additional deformation network. As this case is based on end-point deformations, it cannot adapt to larger changes of surface structure in the middle of the domain. In contrast, using our full pipeline with the deformation network yields surfaces that adapt to the varying behavior in the interior of the parameter space, as shown on the right side of Fig. 5. However, it is also apparent that the deformations generated by our approach do not capture every detail of the references. The solution we retrieve is regularized by the varying reference surfaces in small neighborhoods of $\alpha$, and the networks learns an averaged behavior from the inputs.

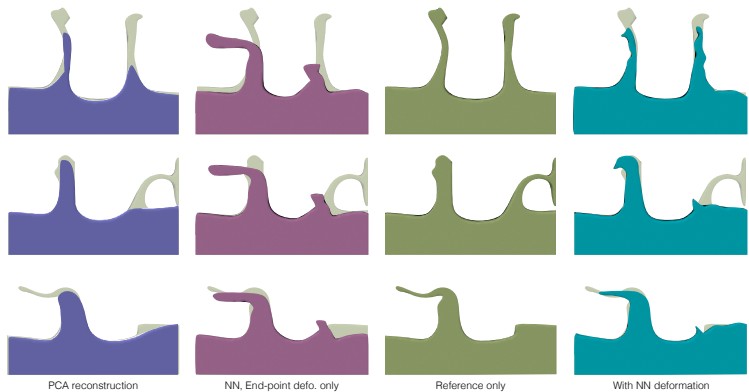

Figure 16: Different example surfaces from the 2D parameter space of Fig. 13. From left to right: surfaces reconstructed with PCA (purple), weighted deformations using a trained parameter network (pink), the reference surfaces (brown), and on the far right the output of our full method with a deformation network (teal). Note that none of the other methods is able to reconstruct both arms of liquid in the first row, as well as the left sheet in the bottom row. The reference surfaces are shown in light brown in the background for each version.

## C.2  4D DATA SETS

Below we give additional details for our results for the 4D data sets and experiments presented in the main document.

**Liquid Drop**   As our first 4D test case, we chose a drop of liquid falling into a basin. As our simulation parameters we chose the $x$- and $y$-coordinates of the initial drop position, as well as the size of the drop. We typically assume that the z-axis points upwards. To generate the training data, we sample the parameter space on a regular grid, and run simulations, each with a spatial resolution of $100^3$ to generate a total of 1764 reference SDFs. Here, $\psi_0$ contains a 4D SDF of a large drop falling into the upper right corner of the basin. In Fig. 17 we show additional examples how the introduction of the deformation network helps to represent the target surface across the full parameter range.

The advantages of our approach also become apparent when comparing our method with a direct interpolation of SDF data-sets, i.e., without any deformation. Our algorithms requires a single full-resolution SDF, three half resolution deformations, and the neural network weights (ca. 53.5k). While a single $40^4$ SDF requires ca. 2.5m scalar values, all deformations and network weights require ca. 2m scalars in total. Thus our representation encodes the full behavior with less storage than two full SDFs. To illustrate this point, we show the result of a direct SDF interpolation in Fig. 18. Here we sample the parameter space with 8 SDFs in total (at all corners of the 3D parameter space). Hence, this version requires more than 4x the storage our approach requires. Despite the additional memory, the direct interpolations of SDFs lead to very obvious, and undesirable artifacts. The results shown on the right side of Fig. 18 neither represent the initial drop in (a), nor the resulting splash in (b). Rather, the SDF interpolation leads to strong ghosting artifacts, and an overall loss of detail. Instead of the single drop and splash that our method produces, it leads to four smoothed, and repeated copies. Both the PCA example above, and this direct SDF interpolation illustrate the usefulness of representing the target surface in terms of a learned deformation.

For the falling drop setup, our video also contains an additional example with a larger number of 14 pre-computed deformations. This illustrates the gains in quality that can be achieved via a larger number of deformation fields. However, in this case the parameter and deformation network only lead to negligible changes in the solution due to the closely matching surface from the pre-computed deformations.

**Stairs**   Our second test setup illustrates a different parameter space that captures a variety of obstacle boundary conditions parametrized with $\alpha$. Our first two simulation parameters are the heights of

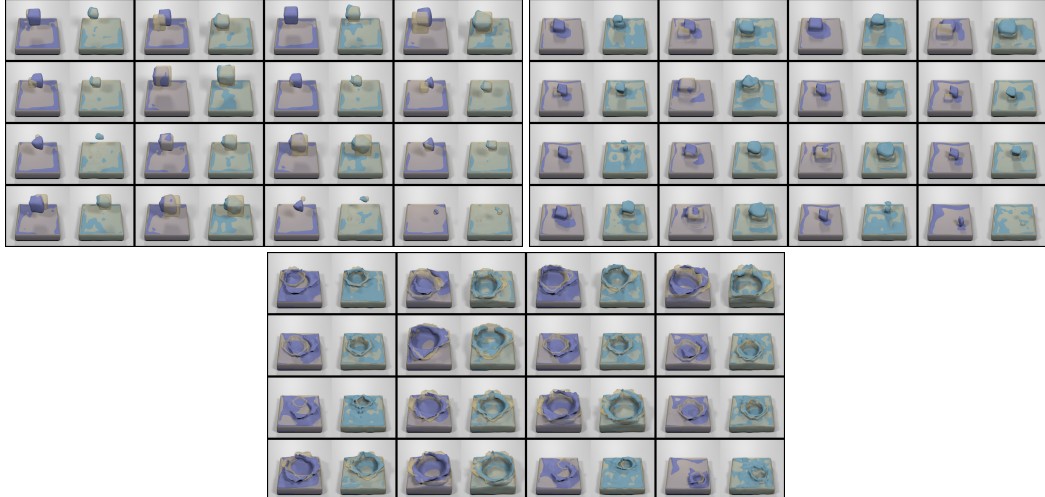

Figure 17: Additional examples of the influence of the deformation network for three different time steps ($t = 1, 4, 8$ from top to bottom). Each pair shows the reference surface in transparent brown, and in purple on the left the deformed surface after applying the precomputed deformations. These surfaces often significantly deviate from the brown target, i.e. the visible purple regions indicates misalignments. In cyan on the right, our final surfaces based on the inferred deformation field. These deformed surface often match the target surface much more closely.

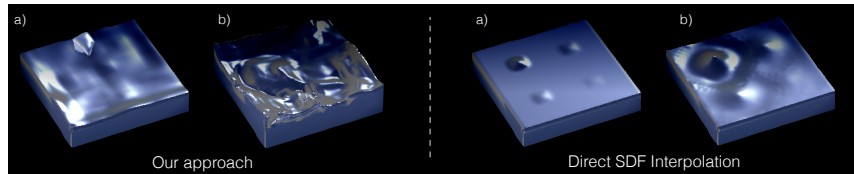

Figure 18: Two frames generated with our approach (left) and with a direct SDF interpolation using a similar amount of overall memory (right). The latter looses the inital drop shape (a), and removes all splash detail (b). In addition, the direct SDF interpolation leads to strong ghosting artifacts with four repeated patterns.

two stair-like steps, while the third parameter is controlling the position of a middle divider obstacle, as illustrated in Fig. 19. The liquid flows in a U-shaped manner around the divider, down the steps. For this setup, we use a higher overall resolution for both space-time SDFs, as well as for the output of the deformation network. Performance details can be found in Table 1.

Fig. 20 depicts still frames captured from our mobile application for this setup. With this setup the user can adjust stair heights and wall width dynamically, while deformations are computed in the background. While this setup has more temporal coherence in its motion than the drop setup, the changing obstacle boundary conditions lead to strongly differing streams over the steps of the obstacle geometry. E.g., changing the position of the divider changes the flow from a narrow, fast stream to a slow, broad front.

Table 1: Performance and setup details of our 4D data sets in the Android app measured on a Samsung S8 device. The *"defo. align"* step contains alignment and rescaling of the deformations.

|        | SDF res. | Defo. res. | NN eval. | Defo. align | Rendering |
|--------|----------|------------|----------|-------------|-----------|
| Drop   | $40^4$   | $20^4$     | 69ms     | 21.5ms      | 21ms      |
| Staris | $50^4$   | $25^4$     | 410ms    | 70ms        | 35ms      |

Table 2: Overview of our 2D and 4D simulation and machine learning setups. Timings were measured on a Xeon E5-1630 with 3.7GHz. *Res*, *SDF* and *Defo* denote resolutions for simulation, training, and the NN deformation, respectively; *Sim* and *Train* denote simulation and training runtimes. $s_p, s_d, \gamma_1, \gamma_2$ denote training steps for parameters, training steps for deformation, and regularization parameters, respectively.

| Setup | Res. | SDF | Defo. | Sim. | Train | $s_p$ | $s_d$ |
|---|---|---|---|---|---|---|---|
| 2D setup, Fig. 13 | $100^2$ | $100^2$ | $25^2$ | - | 186s | 40k | 10k |
| Drop, Fig. 8 | $100^3 \cdot 100$ | $40^4$ | $10^4$ | 8.8m | 22m | 12k | 2k |
| Stairs, Fig. 20 | $110^3 \cdot 110$ | $50^4$ | $15^4$ | 9.7m | 186m | 9k | 1k |

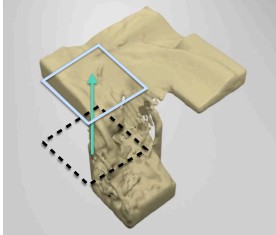 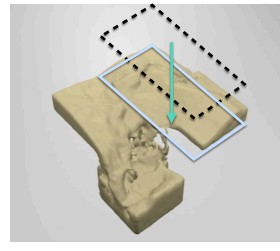 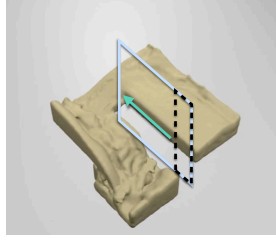

Parameter 1 - raise corner    Parameter 2 - lower platform    Parameter 3 - wall width

Figure 19: The geometric setup of the three deformations of our stairs setup from 20 are illustrated in this figure.

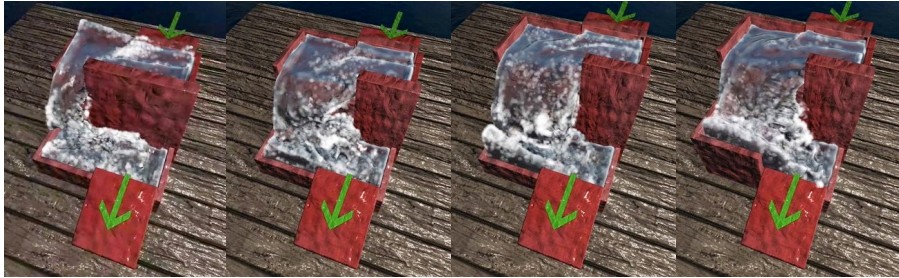

Figure 20: These screens illustrate our stairs setup running in our mobile application. From left to right, the middle divider is pulled back, leading to an increased flow over the step in the back. In the right-most image, the left corner starts to move up, leading to a new stream of liquid pouring down into the outflow region in the right corner of the simulation domain.

