# OpenReview forum: "Generating Liquid Simulations with Deformation-aware Neural Networks"
_ICLR.cc/2019/Conference_

### Official Review · AnonReviewer3 · 2018-11-02
**Good applied paper appropriate for a workshop**

**Rating:** 5
**Confidence:** 4

**Review:**

The paper presents a coupled deep learning approach for generating realistic liquid simulation data that can be useful for real-time decision support applications. While this is a good applied paper with a large variety of experimental results, there is a significant lack of novelty from a machine learning perspective.

1. The primary novelty here is in the problem formulation (e.g., defining cost function etc.) where two networks are used, one for learning appropriate deformation parameters and the other to generate the actual liquid shapes. This is an interesting idea to generate the required training data and build a generalizable model.

2. But based on my understanding, this does not really explicitly incorporate the physical laws within the learning model and can't guarantee that the generated data would obey the physical laws and invariances. So, this is closer to a graphics approach and deep learning has been used before extensively in a similar manner for shape generation, shape transformation etc.

3. In terms of practical applications, to the best of my knowledge there are sophisticated physics-based and graphics based approaches that perform very fast fluid simulations. So, the authors need to provide accuracy and computation cost/time comparisons with such methods to establish the benefits of using a deep learning based surrogate model.

xxxxxxxxxxxxxxxxxxx

I appreciate the rebuttals from the authors, updated my score, but I still believe (just like another reviewer) that this is better suited for a workshop or a conference like SIGGRAPH.

---

> ### Author Response · Authors · 2018-11-19
> **Thanks for your review.**
>
> Thank you for the valuable comments which are addressed as follows.
>
> “...does not really explicitly incorporate the physical laws within the learning model…”
>
> We agree, this would be an interesting direction for future work. We have focused on a more generic method that could also applied to other areas with different or no physical constraints. And for all such extensions it is important to establish how well the core of the method works, which we have focus on in our submission.
>
> “...sophisticated physics-based and graphics based approaches that perform very fast fluid simulations…”
>
> Yes, this is a very good point, and we believe the performance of our trained models is the main strength of our method. We have a variety of solvers available in our group, and we have long standing experience with high-performance of fluid solvers. We can confidently say that no other method currently comes close in terms of performance. Our cell phone implementation is a good indicator of this: despite being published in early 2017, no other 3D liquid simulations is available on Android so far (to the best of our knowledge).
>
> “... provide accuracy and computation cost/time comparisons with such methods …”
>
> As we mention in the paper, our method is currently more than three orders of magnitude faster than a reference Eulerian CPU-based solver. We’d be happy to add a comparison where the CPU solver accuracy is reduced to levels matching our deformation learning algorithm (cf. Fig. 4). We have previously also performed tests with SPH-based solvers on Android smartphones. However, despite GPU-optimizations we were only able to achieve simulations with less than 10k particles, which led to smooth simulations with very few details. Thus, for a given computational budget, neither Lagrangian nor Eulerian simulations methods can currently give a quality similar to our deep learning model.

---

### Official Review · AnonReviewer1 · 2018-11-03
**solid applications paper**

**Rating:** 7
**Confidence:** 4

**Review:**

This is an application paper on dense volumetric synthesis of liquids and smoke. Given densely registered 4D implicit surfaces (volumes over time) for a structured scene, a neural-network based model is used to interpolate simulations for novel scene conditions (e.g. position and size of dropped water ball). The interpolation model composes two components -- given these conditions, it first regresses weights combining a set of precomputed deformation fields, and then a second model regresses dense volumetric deformation corrections -- these are helpful as some events are not easily modeled with a set of basis deformations.

I found the paper hard to read at first, since the paper is heavy on terminology, only really understood what is going on when I went through the examples in the appendix, which are helpful and then on a second read the content was clear and appears technically correct. I would advise considering defining in more detail early the problem setup (e.g. Fig 13 was helpful), explain some of the variables in context.

This is primarily an application paper on simulating liquids in controlled scenes using nets and appears novel in that narrow domain. The specific way deformations are composed -- using v_inv to backwards correct basis deformations, following up the mixing of those with a correction model -- is intuitive and is also something I see for the first time.

The experimental results are sufficient for simulating liquids/smoke, except I would like to also see a comparison to using deformation field network only, without its predecessor. This was done for Fig 6, but would be nice to also see it numerically in ablation in Fig. 4. Another useful experiment would be to vary the number of bases and/or the resolution of the deformation correction network and see the effects.

More importantly, it would be very helpful is to try this approach for modeling deforming object and body shapes for which there are many datasets (e.g. Shapenet). Right now the implicit surface deformation model is only tested on liquids examples, which limits the impact to that specialist domain -- it's a bit more of a SIGGRAPH type of paper than ICLR.

---- Post author feedback comment ----
I raised my rating to 7 as the paper itself is solid, main concern as another reviewer points out is it may be a bit too specialist for ICLR. If the AC decides to reject based on this fact I am ok with that as well.

I think it would be helpful to add more ablation (deformation-only results for all cases) and experiments with different numbers of bases in the final version. If that's added it will strengthen the paper.

---

> ### Author Response · Authors · 2018-11-19
> **Thanks for your review.**
>
> Thank you for the helpful feedback.
>
> - Regarding “... would be nice to also see it numerically as ablation in Fig. 4 …”:
>
> Actually, our examples contain an ablation study, although we agree that it could be more clearly presented as such. There is an entry for a version without parameter learning in Fig. 4, which we refer to as ‘flat’, indicating that there is only a as-simple-as possible initial shape, and no pre-computed deformation. Thus, an ablation study is given for the drop setup with the versions: 1) initial error, 2) parameter learning only without deformation learning, 3) deformation learning only (‘flat’ example), 4) full method. We will revise our document accordingly to clarify this.
>
> - Regarding “...  trying this approach for modeling deforming object and body shapes … “:
>
> This would be very interesting. In our case, we have focused on fluid simulations because they are already a difficult problem, but another test case would help to prove the generalizability of our approach.

---

> > ### Comment · AnonReviewer1 · 2018-11-26
> > **Flat example**
> >
> > Thank you for clarifying. I saw that but it would help to have deformation-learning-only column in Fig 4, since "flat" case is a bit of a corner case.

---

### Official Review · AnonReviewer2 · 2018-11-03
**A Good Attempt towards Applying Deep Learning for Physical Simulation**

**Rating:** 7
**Confidence:** 3

**Review:**

This paper introduces a deep learning approach for physical simulation. The approach combines two networks for synthesizing 4D data that represents 3D physical simulations. Here the first network outputs an initial guess, and the second network adds details. The first network utilizes a set of precomputed deformations, while the weights can be set to generate different output shapes. The precomputed deformations are applied in a recurrent manner. The second network is a variant of STN.

The results are impressive from the perspective of the current abilities of deep neural networks. The synthesized simulations are not physically accurate, but with certain visual realism. Experimental results are sufficient.

However, it is also necessarily to add more intuitions to the current approach. First, it would be good to discuss why the current network design is desired. For example, when designing the first network, can we also design another neural network that applies the deformation backwards and enforce some consistency to improve the results? Also, many simulations use adaptive sampling (high-resolution near the surface and low-residual in the interior). Can we use an adaptive grid-structure (say Octree) to increase the resolution?

Also, is there a simple setting so that the current network design generates accurate results. If not, would increase the number of pre-computed deformations improve the approximation. If so, what would be the optimal basis for $u_i$? What is the tradeoff between using more basis for the first network and increasing the complexity of the second network?

For visualization, it would also good to show the 3D grid.

Overall, it is good paper to see at ICLR.

---

> ### Author Response · Authors · 2018-11-19
> **Thanks for your review.**
>
> Thank you for the detailed suggestions and encouraging comments.
>
> - Regarding “... applying the deformation backwards to enforce consistency …”:
>
> This is an interesting direction, checking consistency between forward and backward steps could yield an estimate of, e.g., loss of momentum. However, as a first additional constraint, we would target divergence freeness, i.e., conservation of mass.
>
> - Regarding “... adaptive grid-structure (say Octree) to increase the resolution …“:
>
> That is a good direction. A focus on the surface would be particularly useful for liquids, and we believe our approach for learning deformations would transfer nicely to adaptive representations like octrees. However, we were not able to try this up to now.
>
> - Regarding “... increase the number of pre-computed deformations to improve the approximation ...”:
>
> Yes, it would improve the approximation quality. We increased the deformations in one of the tests and obtained correspondingly better results. The size of the basis is a compromise between memory/pre-computing time and quality. In our case, we have reduced the basis as much as possible, taking into account the degrees of freedom of interaction. If the reviewers wish, we could add the example with additional deformations to our submission.

---

### Meta-Review · Area_Chair1 · 2018-12-16
**solid applications paper;   final decision is subjective;  borderline**

**Confidence:** 2
**Recommendation:** Accept (Poster)

**Metareview:**

This paper presents a novel method for synthesizing fluid simulations, constrained to a set of parameterized variations,
such as the size and position of a water ball that is dropped. The results are solid; there is little related
work to compare to, in terms of methods that can "compute"/recall simulations at that speed.
The method is 2000x faster than the orginal simulations. This comes with the caveats that:
(a) the results are specific to the given set of parameterized environments; the method is learning a
compressed version of the original animations; (b) there is a loss of accuracy, and therefore
also a loss of visual plausibility.

The AC notes that the paper should use the ICLR format for citations, i.e., "(foo et al.)" rather than "(19)".
The AC also suggests that limitations should also be clearly documented, i.e., as seen from the
perspective of those working in the fluid simulation domain.

The principle (and only?) contentious issue relates to the suitability of the paper for the ICLR audience,
given its focus on the specific domain of fluid simulations.  The AC is of two minds on this:
(i) the fluid simulation domain has different characteristics to other domains, and thu
understanding the ICLR audience can benefit from the specific nature of the predictive problems that
come the fluid simulation domain;  new problems can drive new methods.  There is a loose connection
between the given work and residual nets, and of course res-nets have also been recently reconceptualized as PDEs.
(ii) it's not clear how much the ICLR audience will get out of the specific solutions being described;
it requires understanding spatial transformer networks and a number of other domain-specific issues.
A problem with this type of paper in terms of graphics/SIGGRAPH is that it can also be seen as "falling short"
there, simply because it is not yet competitive in terms of visual quality or the generality of
fluid simulators;  it really fulfills a different niche than classical fluid simulators.

The AC leans slightly in favor of acceptance, but is otherwise on the fence.